# Adult Animal Stem Cell-Derived Organoids in Biomedical Research and the One Health Paradigm

**DOI:** 10.3390/ijms25020701

**Published:** 2024-01-05

**Authors:** Vojtech Gabriel, Christopher Zdyrski, Dipak K. Sahoo, Abigail Ralston, Hannah Wickham, Agnes Bourgois-Mochel, Basant Ahmed, Maria M. Merodio, Karel Paukner, Pablo Piñeyro, Jamie Kopper, Eric W. Rowe, Jodi D. Smith, David Meyerholz, Amir Kol, Austin Viall, Mohamed Elbadawy, Jonathan P. Mochel, Karin Allenspach

**Affiliations:** 1Department of Biomedical Sciences, College of Veterinary Medicine, Iowa State University, Ames, IA 50011, USA; hannah.marie.wickham@gmail.com (H.W.); atbasant@iastate.edu (B.A.); jpmochel@uga.edu (J.P.M.); 23D Health Solutions Inc., Ames, IA 50010, USA; christopher.zdyrski@3dhealth.solutions (C.Z.); ralston.abigail.l@gmail.com (A.R.); mmaria@iastate.edu (M.M.M.); 3Department of Veterinary Clinical Sciences, Iowa State University, Ames, IA 50011, USA; dsahoo@iastate.edu (D.K.S.); abmochel@iastate.edu (A.B.-M.); jkopper@iastate.edu (J.K.); 4Atherosclerosis Research Laboratory, Centre for Experimental Medicine, Institute for Clinical and Experimental Medicine, 14021 Prague, Czech Republic; pauknerkarel@gmail.com; 5Department of Veterinary Diagnostic and Production Animal Medicine, College of Veterinary Medicine, Iowa State University, Ames, IA 50011, USA; pablop@iastate.edu (P.P.); jdismith@iastate.edu (J.D.S.); 6Department of Pathology, University of Iowa, Iowa City, IA 52242, USA; david-meyerholz@uiowa.edu; 7Department of Pathology, University of California, Davis, CA 94143, USA; akol@ucdavis.edu (A.K.); akviall@ucdavis.edu (A.V.); 8Department of Pathology, College of Veterinary Medicine, University of Georgia, Athens, GA 30530, USA; me02824@uga.edu; 9Department of Pharmacology, Faculty of Veterinary Medicine, Benha University, Toukh 13736, Egypt

**Keywords:** organoid, biomedical, reverse translational research, veterinary medicine

## Abstract

Preclinical biomedical research is limited by the predictiveness of in vivo and in vitro models. While in vivo models offer the most complex system for experimentation, they are also limited by ethical, financial, and experimental constraints. In vitro models are simplified models that do not offer the same complexity as living animals but do offer financial affordability and more experimental freedom; therefore, they are commonly used. Traditional 2D cell lines cannot fully simulate the complexity of the epithelium of healthy organs and limit scientific progress. The One Health Initiative was established to consolidate human, animal, and environmental health while also tackling complex and multifactorial medical problems. Reverse translational research allows for the sharing of knowledge between clinical research in veterinary and human medicine. Recently, organoid technology has been developed to mimic the original organ’s epithelial microstructure and function more reliably. While human and murine organoids are available, numerous other organoids have been derived from traditional veterinary animals and exotic species in the last decade. With these additional organoid models, species previously excluded from in vitro research are becoming accessible, therefore unlocking potential translational and reverse translational applications of animals with unique adaptations that overcome common problems in veterinary and human medicine.

## 1. Introduction

Translational research is a term anchored in the 1990s, but this traditional approach started to be formed over half a century earlier [1]. This concept promotes employing animals as human health and disease models in biomedical and pharmacological research. Rodents, dogs, nonhuman primates, and other animals are commonly used to model human diseases, with the intention that these data will eventually be applied (translated) to human medicine. While translational research is now a common driving force in biomedical research, this approach is generally objected to due to concerns about experimental animal health, their pain perception, and the animals’ overall welfare [2]. Numerous overseeing committees have been established to create and enforce rules of animal experimental conduct and supervise animal ethical treatment [3]. While this effort improves research animal welfare, some issues still need to be addressed, including the expansion of minimum care requirements, easier accessibility to laboratory plans and reports, and continuous education on the topic [4].

The principles of the 3Rs (Replacement, Reduction, and Refinement) have been a mantra at the forefront of experimental animal research [5]. The Replacement principle promotes the employment of alternative in vitro or in silico nonanimal models, and the Reduction principle refers to the decrease in animal numbers in experimental settings. Finally, the Refinement principle relates to the decrease in animal suffering and an increase in animal welfare in research conditions. Ethical concerns are not the only challenge during translational research. Pharmacological efficacy and toxicity trials often fail to translate the results of the tested substance from animal models to humans [6]. Rodents are the most common biomedical model in pharmacological research and can be effectively used in monogenetic disease investigation [7]. However, the translational potential of the model for investigating chronic multifactorial diseases has been questioned [8].

Rodents often do not develop the chronic multifactorial diseases being investigated using these models; therefore, a nonetiological insult to the organ must be performed to mimic the disease. An example of this approach is models of nonalcoholic steatohepatitis (NASH) [9], which is a typical multifactorial disease influenced by genetic, dietary, environmental, and lifestyle factors that do not normally occur in rodents. Therefore, according to the induction of hepatic injury, rodent models for NASH research must be divided into three groups (dietary, genetic, and chemical). (1) The dietary model provides input on the nutritional component but omits genetic and lifestyle factors. (2) The genetic model can be enhanced by the induction of dietary changes but will still omit other lifestyle factors. Finally, (3) the chemical model can include a high-fat diet (dietary factor) but will not be influenced by genetic or other lifestyle factors [10]. Therefore, rodents cannot offer a complex model of multifactorial disease that do not naturally occur in their species, and the scientific community needs to embrace novel approaches to investigate these diseases.

The One Health Initiative is a multidisciplinary unifying approach to balance and optimize human, animal, and environmental health. While the term was coined in 2004 [11], the initial idea began forming in the early 19th century [12]. The approach promotes knowledge sharing between human and veterinary medicine, and its main aim is to stimulate biomedical research, improve clinical and paraclinical care, find solutions to biosafety issues, and increase public health efficacy [13]. The One Health Initiative is visually summarized in Figure 1A. The approach is further applied in the One Medicine approach.

The One Medicine approach, derived from the One Health Initiative, perceives veterinary and human medicine as parts of one discipline. Therefore, clinical knowledge should be shared to benefit both human and veterinary patients. The One Medicine approach embraces the concept of reverse translation, which assumes that the data and information from human and veterinary clinical research can be translated, under the right conditions, between the biomedical fields. Veterinary patients suffering from a disease described in the human population can help to improve human patient outcomes by providing data on diagnosis, treatment, or outcomes. Conversely, information on diseases better explored in human medicine can be used to improve knowledge in veterinary medicine [14]. The diagram representing the One Medicine approach and reverse translational research is presented in Figure 1B.

Reverse translational research can provide important advantages compared to traditional translational models [14]. The most important features are related to animal domestication, as companion animal patients share a similar environment, and very often sedentary lifestyles, with their owners. Furthermore, pet nutrition has also followed trends in human food processing for decades [15]. The rising obesity in humans and companion animals is one example of shared environmental factors [16,17,18]. The combination of reverse translational research and novel in vitro models could provide reliable research tools that are less reliant on sample donors.

Furthermore, personalized and regenerative medicine are becoming more relevant with our expanding knowledge of patient-specific treatment options for both veterinary and human patients. The screening of multiple drug candidates at different concentrations and the patient’s response can be tested in vitro prior to treatment in vivo. Personalized organoids have the potential to rapidly advance this therapeutic option in the medical field. Beyond personalized drug screening, both transplants and the repair of diseased tissue have also been of great interest in the organoid field.

## 2. Brief Overview of Most Common In Vitro Cell Culture Models

### 2.1. Primary Cell Lines

Primary cell cultures can be isolated directly from the patient and employed in a personalized medicine setting. However, using primary epithelial cell lines in biomedical research has specific disadvantages, limiting their use. Some of these difficulties include the isolation of single cell types and the limited life span of the culture [19]. This limited life span of cell lines greatly hampers their use, and therefore, many researchers turn to the use of commercially proven and immortalized cell lines.

### 2.2. 2D Immortalized Cell Lines

Conventional 2D cell cultures have benefits in biomedical research, including their ease of use and low cost of maintenance. Additionally, there is a great deal of prior knowledge and testing for these immortalized cell lines, which have been extensively characterized. Typically, these cell lines are composed of cells that adhere to a flat surface, such as a flask or plate, in which they can proliferate [20]. However, these again come with limitations, such as the alteration of cellular morphology after isolation, limiting the complexity of this cell mode [19]. These conventional 2D cell lines are commonly used as a first model in basic and applied research but typically lack the structural and functional complexity of in vivo tissues, hampering their use as informative or personalized biomedical models.

Traditional 2D cell lines are primarily of cancerous origin and remain the preferred strategy for drug in vitro intestinal permeability assays. However, these cellular monolayers cannot accurately simulate the physiology of real organs [21]. The Caco-2 cell line was developed in 1977 from human colorectal adenocarcinoma cells and is the “gold standard” for intestinal permeability assays [22]. While experiments with Caco-2 cells provide some intestinal permeability data, this approach also has specific limitations. Caco-2 represents one changed cell type and does not provide the much-needed variety of cells in a healthy intestine (goblet cell, enteroendocrine cells) [23]. Furthermore, Caco-2 cell permeability is decreased due to altered tight junction function compared to intestinal tissue [21]. Cytochrome P450 3A4 (CYP3A4), a metabolic enzyme responsible for xenobiotic oxidation, is also low or absent in a Caco-2 culture [24]. The same problem can be observed with the xenobiotic efflux pump P-glycoprotein [25] (P-GP), xenobiotic nuclear pregnane X [26], and steroid X receptors [27] (PXR and SXR). The lack of proper cellular machinery limits the use of this cancerous cell line and does not correspond to the situation observed in vivo [24]. With the inherent constraints of 2D cell cultures in accurately mimicking the in vivo environment, a model with increased complexity and cell–cell interactions can further build upon research performed in 2D cultures.

### 2.3. Stem Cell-Derived Organoids

Organoids are three-dimensional (3D) stem cell-derived multicellular in vitro constructs that can reliably mimic the microanatomy and physiology of their corresponding organ [28,29]. The organoid technology can be used in numerous fields, from developmental biology, pathophysiology, drug discovery and testing, and toxicology to studying infectious diseases and regenerative medicine [30] (Figure 2).

Organoids can be derived from either pluripotent or adult stem cells. Pluripotent stem cells can be obtained through an embryonic stem cell (ESC) harvest or derived from reprogrammed adult somatic cells termed induced pluripotent stem cells (iPSCs). Unlike adult stem cell-derived (AdSC) organoids, ESC and iPSC-derived structures often resemble embryonic or neonatal cells, lack full maturation, and do not perform the same tasks as adult cells in vivo [31]. However, pluripotent stem cell-derived organoids serve as an excellent model for the study of embryogenesis and tissue development. Another advantage of this type of organoid is that its cells encompass every tissue layer represented in vivo.

AdSC-derived organoids represent a simplified version of iPSC-derived organoids. They consist mainly of epithelial cell types of endodermal origin [32], but they can be generated without the need for iPSC reprogramming [33]. While AdSC-derived organoids are useful for mimicking certain organs (endodermal origin), their use for ectodermal and mesodermal-derived structures is limited. Unlike AdSC-derived organoids, iPSC-derived organoids have broad implications for research in areas of cardiology, neurology, and bone disease. AdSC-derived organoids offer indefinite proliferation of the 3D culture with the possibility to bioarchive the samples [30]. Finally, these organoids do not require embryonic cells and can be established from biopsies (fine-needle, Tru-cut, or excisional samples), fulfilling the needs of the reverse translational approach.

Organoids were first derived from the adult murine intestine and described by Sato et al. in 2009 [29] following the discovery of an intestinal multipotent stem cell marker, leucine-rich repeat-containing G-protein-coupled receptor 5 (Lgr5), by Barker et al. in 2007 [34]. Sato et al. used cell sorting to select for Lgr5^+^ cells and established the basis of organoid expansion media including components such as R-spondin 1 (Wnt pathway agonist), EGF (epidermal growth factor), and Noggin (bone morphogenic protein modulator). Finally, multiple extracellular membrane matrices are available; however, Matrigel, a solubilized basement membrane matrix, has provided a favorable 3D environment for organoid proliferation and differentiation. These techniques, with some modifications, have become the basis of modern AdSC-derived organoid protocols. The isolation process for other tissues have been similar with possible other markers used to detect stem cells (e.g., EpCAM for liver) [35]. Additionally, Zdyrski et al. reported the isolation of various canine tissues using identical procedures leading to standardization of these techniques [36]. 

Organoid lines from numerous species have been successfully derived, including members of the classes Aves [37] and Reptilia [38]; however, the current emphasis has been on mammalian organoids. This text will further focus exclusively on AdSC-derived organoids due to the span of the field. Organoids derived from humans and rodents will be briefly introduced as a basis for the general organoid development timeline. Furthermore, the introduction will be mostly centered on healthy animal-derived organoids from exotic and traditional veterinary species and their use in reverse translation research. Figure 3 summarizes the timeline of human and animal organoid development.

## 3. Reproducibility and Data Deposition

The organoid (stem cell) research field is an ever-changing novel area of biomedical research. Progress is hindered by the lack of standardization (especially in veterinary medicine). While there are efforts to establish standardized culturing protocols for veterinary organoid cultures [30], the need for reproducible data and methods is clear. The characterization of organoid markers, differences between species and individuals, and the developmental stage of the organoids themselves have to be strictly defined to establish baselines for the field [39]. Another concern regarding reproducibility are the hydrogels that are used for stem cell cultures; very often, these have animal-derived undefined components and can vary batch to batch [40]. Data deposition is also a major problem and the culture of free data sharing promoted in the scientific world is progressing slowly forward. The inability to compare data directly is slowing down research efforts and leaves experimental data out of reach for review by the broad scientific community. 

The most recent PubMed search was performed on 14 November 2022. There were 14,233 results for organoid* AND (human OR murine OR canine OR equine OR feline OR horse OR Birds* OR dogs OR cats). A similar research method was utilized for the review with organoid* AND (canine OR equine OR feline OR horse* OR dogs OR cats), yielding 460 results; an additional selection based on abstracts and content yielded a very comparable list of papers.

Moreover, due to the importance of the stem cell marker (LGR5) and Wnt in the development of adult epithelial stem cells [29], we confined our selection to adult stem cell-derived organoids dependent on stem cell markers or Wnt signaling. Additionally, the search was limited to the first occurrence of organ and animal organoid line development (see Table 1).

### 3.1. Human Organoids

Jung et al. described a protocol for isolating and expanding human colon-derived organoids (colonoids) from adult stem cells in 2011 [70]. This system was further modified in the following decade to generate human organoids derived from the prostate [71], proximal airways [72], liver [73], pancreas [74], stomach [75], distal airways [76], fallopian tube [77], salivary gland [78], placenta [79], urinary bladder [80], spleen [81], mammary gland [82], nasal epithelium [83], and thyroid gland [84].

While human-derived organoids have the potential to mimic human in vivo environments precisely, there are also several setbacks to the technology. The number of donors is limited, and harvesting healthy tissue is ethically questionable. Ethical challenges have been not limited to the organoid technology and also include its combination with other techniques, such as creating chimeras, organoid gene editing, organoid transplantation trials, or simply commercializing human-derived technology [85]. The clear limitations in in vivo experimentation on donors further hinder pathophysiological or pharmaceutical studies. 

### 3.2. Murine Organoids

Rodents are a commonly used model in biomedical research, with mice being the most used species, and numerous genetically modified types being available. Furthermore, due to their popularity, relatively inexpensive upkeep, and high reproduction rate, they are one of the most affordable experimental animal models [7]. The long-standing scientific familiarity with donor animal physiology and a comparison with previous in vivo experiments make murine organoids a good choice for translational research.

Additional murine organoids representing other organs were established soon after the first murine intestinal organoids were created by Sato et al. [29]. Mouse-derived organoids usually represent the primary focus of developing organoid technology followed by the production of human organoids. The list of described murine organoids includes the proximal airways [86], stomach [87], pancreas [88], liver [89], distal airways [90], tongue [91], esophagus [92], prostate [71], salivary gland [93], mammary gland [94], fallopian tube [95], thyroid gland [96], urinary bladder [97], epididymis [98], and spleen [81].

While mice are a popular model due to their low cost and readily available genetic lines [7], the model also has disadvantages (for example, their shorter lifespan prevents the onset of chronic diseases or study of non-naturally occurring diseases) that can limit the translational potential of this model [8]. While progress in murine organoids in terms of translational research is the right decision for the welfare of rodents, the potential for the reverse translational potential in this species is limited.

### 3.3. Other Animal-Derived Organoids

In the last decade, numerous additional species (exotic or traditional veterinary animals) were successfully utilized, and their cells supported organoid culture growth based on the modification of human and murine protocols [99]. The successfully established cell lines typically served as proof-of-concept studies for applying animal-derived organoids in numerous biomedical areas [23].

In 2017, Powell et al. investigated why organoids can be established easily from different species or classes of the animal kingdom [56]. The authors found broad protein sequence conservation when comparing the three major growth factors (Wnt3a, R-spondin, and Noggin) between humans, rodents, and other animals, including dogs, cats, cows, horses, pigs, goats, sheep, and chickens. This high genetic similarity allows murine/human growth factors to stimulate organoid expansion and growth in nonhuman/nonmurine species.

Organoids derived from exotic/nontraditional animal species can be important for studying the original species’ unique physiologies and environments or disease pathogenesis [64]. While, for the most part, these cell lines cannot be used in reverse translational research, they further promote the goals of the One Health Initiative.

However, organoids derived from highly explored veterinary species likely have the best potential for reverse translational research applications [14]. Medical knowledge about and research performed on the species are prerequisites. Furthermore, domestication’s influence via strong phenotypic selection and constant interaction with humans is also crucial [100]. Domestication changes the environment and dietary habits of animals to be compatible with the human lifestyle [101]. In 2021, Beaumont et al. released a review that summarized the development of organoid models derived from pigs, ruminants, equines, and chickens [102]. The manuscript offers information on individual literature publications and summarizes organoid isolation methods and culture media composition [102].

The specific category of veterinary species that are companion animals (pet animals) also share the same household (environment) with their owners; their nutrition is markedly changed to correspond to human food, and they often share the same habits as their owners, for example, a sedentary lifestyle [103]. The most promising potential for reverse translational research implementation is currently canine organoids [14,23]. The results gained from drug clinical trials on dogs are still considered superior to those on rodents by the FDA, even in a translational research setting. Canine medicine is one of the most advanced research fields in veterinary medicine and generates a large amount of data that could be translated into human medicine. These factors make canines an ideal target organism for reverse translational research [14]. Canine organoid technology has evolved rapidly in the last decade. The main methods used to process intestinal and hepatic organoids have been published, and novel organoid lines have been described [23,66]. Here, we discuss possible applications of several organoid culture systems, focusing on research opportunities stemming from animal-derived organoids.

## 4. Main Research Applications

### 4.1. Intestinal Organoids

Intestinal organoids are a valuable tool for investigating intestinal physiology in numerous species. Monkey intestinal organoids originating from the duodenum, jejunum, ileum, cecum, and colon were established by Inaba et al. in 2021 [47]. A rhesus monkey (*Macaca mulatta*) and a Japanese macaque (*Macaca fuscata*) were used to establish the culture, and the group reported continuous growth of the organoids for six months. The authors followed the murine intestinal organoid establishment protocol by Mahe et al. in 2013 [104]. 

In 2021, Li et al. reported the establishment of colonic organoids derived from two cynomolgus monkeys, also called crab-eating monkeys (*Macaca fascicularis*) [44]. The protocol for human colonic stem cell isolation published by Jung et al. in 2011 was used to establish the culture [44]. The reduction in Wnt signaling via a decrease in CHIR-99021 (GSK-3 inhibitor), and withdrawal of prostaglandin E2 (PGE2), were needed for the final differentiation of the culture. The authors noted that prolonged organoid culture in these conditions led to the death of the cultured cells. Li et al. kept organoids in differentiation media for one week and harvested the culture for analysis. They concluded that the medium could properly differentiate enterocytes and goblet cells but was less effective for enteroendocrine cell differentiation. The developmental stage and functional properties of these organoids have to be further investigated. Similarly, as in humans, the withdrawal of Wnt3a is necessary for cell differentiation and maturation but also causes a collapse of the macaque intestinal organoid culture. SB202190 (P38 MAPK inhibitor) and nicotinamide were also removed from the media to support differentiation. Inaba et al. investigated intestinal taste-like chemosensory cells (enteroendocrine and tuft cells) and achieved their differentiation in culture with the addition of interleukin 4 (IL-4) and the Notch inhibitor dibenzazepine (DBZ). The authors analyzed organoid expression and chemoreception reactions and concluded that monkey-derived taste-like chemosensory cells are prone to be closer in function to those in humans than to those in rodents. In another publication authored by Inaba et al. in 2021 [105], the authors improved monkey intestinal culture methods via the removal of SB202190 (P38 MAPK inhibitor) and its replacement with insulin-like growth factor 1 (IGF-1) and basic fibroblast growth factor (FGF-2), following a method described by Fujii et al. in 2018 for human intestinal organoids [106]. This change to the expansion media significantly increased the percentage of organoids larger than 150 µm from 25% to 60% in the culture the second day after passage or isolation. Furthermore, the addition of IL-4 and IL-13 induced the expression of tuft, goblet, and Paneth cell markers while decreasing the expression of the stem cell marker LGR5. These substances also enhanced the production and accumulation of acetylcholine in the tuft cells of the macaque intestinal organoids.

The first galline jejunal organoids were isolated in 2018 by Li et al. [48] based on modified protocols for murine intestinal organoids published by von Furstenberg et al. in 2011 [54], Wang et al. in 2013 [107], and Ahmad et al. in 2014 [108]. Fifteen Hylene chickens were used as jejunal donors, and they were divided into three groups to test different concentrations of EGF, Noggin, and R-Spondin-1. The largest diameter of randomly selected measured organoids was in a group using 50 ng/mL EGF, 100 ng/mL Noggin, and 500 ng/mL R-spondin 1. Furthermore, the process of centrifuging the sample down to a pellet had to be adjusted. The authors experimented with a centripetal force of 35× *g* for 2 min but found villi in the pellet. The force was decreased to 25× *g*, which led to the detection of a suspension of cells, clusters, and crypts in the pellet; this force was further used to establish the organoid lines.

A novel approach to galline small intestinal organoid (duodenum and jejunum) culture was proposed by Acharya et al. in 2020 [58]. The authors avoided using an extracellular matrix as a structural support, reduced the number of growth factors needed to establish the organoid culture, and grew organoids in an apical-out format. The mucosa of fresh tissue was extruded, triturated, and centrifuged. The pellet was resuspended in fetal bovine serum (FBS), insulin transferrin selenite (ITS), polyamine (in vivo these most likely come from the intestinal bacteria), and bovine pituitary extract, and cultivated overnight. This treatment activated the sheared ends of the villi to self-repair and form organoids. Enteroids were subsequently strained and used for downstream applications. The authors noted a trophic effect on organoid culture when supplemented with EGF and IGF-1. While this approach cannot be called a long-term organoid culture, the methods used here might also be applicable to establishing a clean and low-cost extracellular matrix-independent culture systems in the future.

The characterization of the organoids was performed using TEM, showing microvilli in the apical part of the organoids, and RT-PCR for stem cell marker gene detection (OlfM4, Znrf3, Hopx, and LGR5). Neutral red staining was used to show living organoids, and an EdU assay was performed to verify proliferative capabilities. The authors also established the influence of CHIR-99021 (GSK-3 inhibitor) on the culture. A statistically significant increase in the expression of the stem cell marker LGR5 was observed after 4 days of incubation compared to the untreated control.

Colon-derived organoids of bovine origin were established by Töpfer et al. in 2019 [49] based on Sato’s human intestinal organoid protocol from 2011 [32]. The colonoids achieved more than 30 passages without losing their proliferative capabilities. The differentiation of the organoids was supported by adding DAPT for 24 h, and the gene expression showed lower LGR5 (stem cell marker) expression. Higher mucin2, chromogranin A, and CA1 marker expression showed the presence of goblet cells, enteroendocrine cells, and enterocytes, respectively. The authors used bovine organoids for a cryopreservation in situ experiment. The organoids were frozen on a plate without harvesting using cryopreservation media and stored at −150 °C for 24 h. The growth of the culture after defrosting was similar to that of the traditionally stored unfrozen control. Furthermore, in a follow-up cytotoxicity assay using staurosporine as a stimulant, the average toxicity of the molecule was not different between cryopreserved and control samples. Bovine organoids were also tested in bioprinting applications and were used to create 2D monolayers on dual-chamber permeable supports. Further investigation into these types of organoids is warranted.

Equine jejunal organoids were further developed in the same year by Stewart et al. [50] from three geldings and two mares ranging from 5 to 14 years of age. The culture conditions were based on a modified protocol for porcine and murine intestinal organoid establishment published by Sato et al. in 2009 [29], Gonzalez et al. in 2013 [52], and Khalil et al. in 2016 [109]. Immunofluorescence staining confirmed by PCR was further used to characterize the cellular population in the organoids, revealing the presence of enteroendocrine cells (chromogranin A), goblet cells (MUC2), and Paneth cells (Lyz). Furthermore, intestinal stem cells (SOX9), transit-amplifying cells (KI67), and reserve stem cells (HOPX) were detected in the organoids.

The first rabbit cecal organoids were derived by Mussard et al. in 2020 [51]. The authors compared growth media based on pharmacological inhibitors (GSK-3 inhibitor for Wnt pathway activation, and LDN193189, a bone morphogenic pathway inhibitor, without WRN) with organoid media based on products of L-WRN 2D cell culture (producing Wnt3a, R-spondin 3 and Noggin). In both cases, the media also contained SB431542 (TGF-β/ALK inhibitor) and Y27632 (associated Rho kinase inhibitor—ROCKi) for anoikis prevention. The results show that rabbit cecum organoid growth is not strictly Wnt pathway activation dependent, as human organoids are (the activation was supported only by GSK-3 inhibitor addition), and that organoids in this media showed more proliferation. However, a low concentration of L-WRN-conditioned media can lead to the differentiation of the organoid culture. Furthermore, Mussard et al. successfully plated and maintained organoids as a 2D monolayer in a permeable support system. Mussard et al. proposed cecal rabbit organoids as a model of intestinal physiology, especially for nutrition and barrier function. The usefulness of these cultures has to be proved by further investigation. 

The first porcine jejunal enteroids were described by Gonzalez et al. in 2013 [52] to serve as a model for mechanistic studies and translational research. Organoids were isolated from tissues of 6- to 8-week-old Yorkshire cross pigs. The crypt buds were described by day 14 and they were fully formed by day 21. The authors identified SRY-box transcription factor 9-positive, SOX9^+^ (stem/progenitor cell), PCNA^+^ (proliferation), MUC2^+^ (Goblet cells), CgA^+^ (enteroendocrine cells), and SIM+ (enterocyte) populations using immunofluorescence techniques. The culture was passaged at 2 weeks after isolation and cultured for 4.5 months without a decrease in organoid proliferation. Porcine duodenal enteroids were established by Koltes et al. in 2016 [55] based on a modified murine intestinal protocol published by Gracz et al. [110]. The organoids were isolated from four adult pigs and grown for 14 to 16 days. A lipopolysaccharide (LPS) stimulation experiment was performed on the culture to investigate the inflammatory response. The internalization of Toll-like receptor 4 (TLR4) into early endosomes was observed, a part of the process leading to adaptive immune response activation. The authors also described the presence of probable enteroendocrine cells (CgA), tight junction formation, and cellular polarization (Claudin 4). Koltes et al. stated that their model could be used to study inflammation signaling and endotoxin tolerance. Organoids derived from porcine esophageal submucosal glands were derived in 2017 [111]. Two types of organoids were identified in the culture, including hollow/ductal spheroids (characterized by the presence of cytokeratin 7) and solid squamous spheroids (characterized by abundant P63). Both spheroids were assessed as highly proliferative and epidermal growth factor dependent. Furthermore, porcine anorectal organoids were derived by Adegbola et al. in 2017 as a model for perianal fistulizing in Crohn’s disease [53]. While the phenotype of rectal organoids was similar to that of small intestine-derived organoids, the cultures derived from the anal and anorectal transition zones displayed a monolayer phenotype.

The canine intestinal organoid culture was described by Chandra et al. in 2019 [23]. The authors generated canine intestinal organoids (duodenum, jejunum, ileum, and colon) based on a modified version of the human intestinal protocol by Saxena et al. from 2015 [112]. Enteroid and colonoid lines were isolated from twenty-eight healthy dogs and twelve diseased individuals, including nine inflammatory bowel disease (IBD) patients, gastrointestinal stromal tumor (GIST) patients, colorectal adenocarcinoma patients, and lysosomal storage disease model animals (MPS type 1). No major morphological differences were observed between the enteroids and colonoids from both healthy and diseased dogs. Transmission electron microscopy (TEM) was used to closely visualize the differentiation of cells, including the presence of microvilli, neurosecretory granules, adherens junctions, tight junctions, and desmosomes. Jejunal enteroids and tissues were further compared using immunohistochemistry showing the expression of pancytokeratin (epithelial marker), chromogranin A (enteroendocrine cells), and PAS (goblet cells). As expected, vimentin and actin (mesenchymal cell markers) and c-Kit and CD3 (immune cell markers) were not present in organoids but were detected in the tissue. Surprisingly, lysozyme (Paneth cell marker) was not present in organoids or the tissue. The authors further employed RNA in situ hybridization (RNA ISH) to detect mRNA, and its distribution in organoids and tissue. LGR5 (stem cell marker) was identified in the enteroid crypt areas, whereas SOX9 (another stem cell marker) was expressed in both the enteroid villus and crypt areas. As reported earlier, this can be explained by the expression of SOX9 in precursors of secretory lineage-specific enteroendocrine and tuft cells [113].

The presence of Paneth cells was further investigated by markers other than lysozyme as Paneth cells with lysozyme-containing vacuoles were not seen during a histological assessment of canine intestine. Paneth cell markers, ephrin type B receptor 2 (EPHB2), and Frizzled 5 (FZD5) were mainly expressed in enteroid crypt areas, indicating that Paneth-like cells likely reside in the crypt areas of the small intestine in this species. Furthermore, alkaline phosphatase (ALP), a brush border marker, has been identified in the enteroid villus area, and NeuroG3 (enteroendocrine cell marker) was dispersed throughout enteroids. Tuft cells were identified using Doublecortin-Like Kinase 1 (Dclk1). The authors performed a set of functional organoid measurements, including optical metabolic imaging (OMI), showing 2-fold increased metabolic activity of the 7-day-old organoids compared to 3-day-old organoids.

A foreskolin swelling assay was used to check the function of the cystic fibrosis conductance regulator (CFTR), and enteroid swelling was observed after 1, 4, and 24 h. Finally, the authors showed that enteroids could phagocytize exosome-like vesicles from the parasite *Ascaris suum*. The authors noted that the canine intestinal organoids could serve as a model for studies on drug toxicity and permeability.

Kramer et al. described the effects of different culture conditions on canine organoids derived from the duodenum, jejunum, and colon in 2020 [114]. The samples were collected from three dogs euthanized for a nonrelated reason, and one duodenum sample was obtained during a routine endoscopic procedure. The authors described the following three types of media: an expansion medium, a differentiation medium (characterized by the removal of nicotinamide and SB202190—P38 MAPK inhibitor), and a refined medium (characterized by the removal of N2, nicotinamide, and SB202190; then EGF is removed after two days of culture, and IGF1 and FGF2 are added to the culture). The authors quantified mRNA transcripts using qPCR of organoids grown in an expansion media and noted high gene expression of LGR5 (stem cell marker) and low expression of the differentiation markers villin 1 (VIL1; enterocyte), chromogranin A (enteroendocrine cells), and MUC2 (goblet cells).

The authors reported that expansion and refined media-grown organoids do not exhibit necrosis, while organoids grown in differentiation media do. This finding was confirmed using an Annexin V assay. Only expansion and refined media supported the long-term growth of the organoids (25 passages). While goblet cells were not present in expansion media, they were confirmed in both differentiation and refined media by MUC2 expression and PAS staining. Through a comparison with other qPCR markers (NEUROG3 and chromogranin A), the authors show similar differentiation between organoids grown in refined and differentiation media. Refined media generated a fourfold higher number of organoids than expansion media. While the refined media formulation looks promising, there is a need for further experimentation with a higher number of samples.

As an in vitro model for reverse translational research, canine organoids possess the ability to grow in a 2D environment in the form of monolayers [115]. The organoids can be applied to systems already engineered for traditional 2D cell cultures. The canine organoid technology can be paired with a traditional permeable support system for drug permeability and toxicity trials. The permeable support system consists of an apical chamber, that simulates the intestinal lumen and basolateral chamber’s communication with the organism’s internal environment.

Ambrosini et al. reported the successful establishment of a canine intestinal organoid culture in a permeable support system in 2020 [116]. The monolayers were established from canine colonoid cultures derived from three individuals. The polarity of the canine organoids was confirmed via TEM by identifying the apically located brush border. Furthermore, TEM was used to identify mucin granules (goblet cells), tight junctions, and desmosomes. The presence of mucus-like molecules on the apical border was further established by live-cell wheat germ agglutinin (WGA) staining.

Additionally, intestinal barrier integrity was confirmed by immunofluorescence staining for zonula ocludens 1, E-cadherin, and a transepithelial electrical resistance (TEER) assessment of the monolayer. The presence of an important efflux pump, P-gp, was detected on the apical side of the monolayer. The authors can support the detection of P-gp by conducting experiments regarding its activity. Ambrosini et al. demonstrated that canine intestinal cultures could be successfully established and characterized in a permeable dual-chamber system. 

Standardization of the dual-chamber permeable support system for canine organoids was established in 2022 by Sahoo et al. [115]. The authors used a permeable support system to create organoid-derived monolayers applicable in intestinal permeability measurements. They demonstrated the usefulness of the model using drugs from the β-blocker family (propranolol, metoprolol, and atenolol) and compared their results to the same measurements in Caco2 culture. The authors designed the experiment as a proof-of-concept study and show promising preliminary results for a species-specific assessment of drug permeability. 

The next example of the evolution of organoid technology for drug permeability and toxicity testing is its incorporation into a microfluidic system. The microfluidic channels are embedded within organoids, and moving media in the channels simulates blood flow in the organism. The organoids are exposed to the shear stress normally present in vivo. An additional advantage of this technology is the possibility of adding coculture chambers; these modules then provide an opportunity to simulate the interaction of the organoid culture with other organs or the immune system [117]. The next step for this technology would be experimentation with hepatocyte differentiation in a chip. 

### 4.2. Modeling Host-Pathogen Interactions

Host–pathogen interactions is another field of research where animal-derived organoids can be specifically useful. While many groups have developed intestinal organoids to study these interactions, additional organoids derived from the lung or abomasum have been suggested for the same purpose.

The first ileal porcine organoids were isolated with their duodenal and jejunal counterparts in 2017 by Powell and Behnke [56] as a host–pathogen model. Five ileum tissue samples were harvested, and L-WRN medium was used to support the growth of the organoids [118]; the organoids survived for 174 days in culture.

Porcine colonoids were developed in 2019 by Li et al. in pursuit of creating a novel model of porcine epidemic diarrhea virus (PEDV) infection [57]. Ileal and colonic organoids were isolated from the same donor and inoculated with PEDV; both cell lines were infected. While the infection of ileal organoids was nonrestricted, the colonoids showed only restricted infection. The number of genome copies was significantly higher in ileal enteroids than in colonoids. This observation was made in a previous study based on tissue sections from infected piglets, where alterations to tight junctions and adherens junctions were observed in the small intestine but not in the colon [119]. Li et al., therefore, concluded that porcine intestinal organoids retain the specific identity of their intestinal segment.

Powell et al. described the first galline cecal organoids in 2017 cultivated with 50% L-WRN-conditioned media [56]. Six individuals were used to establish the organoid lines with a success rate of 66%, and the organoid’s highest passage number was 35 (representing 125 days of culture). The organoids had low vimentin expression (representing independence in coculture with mesenchymal cells) and significantly higher LGR5 (stem cell/progenitor marker) expression than that in tissue.

The first bovine ileal organoids were described in 2017 and were isolated from five donors [56]. L-WRN medium was used to maintain the growth and expansion of the organoids for 45 passages (165 days). The enteroids expressed LGR5 (stem cell marker) at the same level found in tissue. At the same time, the expression of LGR5 was lacking in traditional bovine-derived cell cultures used for intestinal research (Madin-Darby bovine kidney—MDBK). This cell line was used specifically for its TEER value properties which are similar to the electrical resistance of the intestinal monolayer. The expression of vimentin (mesenchymal marker) was high in MDBK cells, but the expression in bovine ileal organoids was low. This observation suggests that supplementing L-WRN-conditioned media is sustainable for bovine organoids, and no additional mesenchymal cells are needed. The authors propose that the model could be used to investigate the growth of *Cryptosporidium parvum*. This use of the technology has to be further investigated.

Bovine jejunal enteroids were developed in 2019 by Derricott et al. as a model for host–pathogen interactions and disease response [59]. Samples from 20–30-month-old cows were used to establish the organoid culture. The organoids were further characterized by immunofluorescence and proteomics techniques. Bovine organoids differentiated into goblet cells (Mucin 2), enteroendocrine cells (chromogranin A), and adherens junctions (E-cadherin). Furthermore, the presence of f-actin in the apical surface of the organoids suggested epithelial cell polarization. Label-free mass spectrometry results uncovered epithelial-specific proteins related to cell junctions and polarization.

Additionally, intestine-specific enterocytes and stem cell proteins were found in the culture. These techniques detected proteins responsible for the morphogenesis of the villi and mucus components. The characterized bovine organoids were subsequently infected with *Toxoplasma gondii* for 1 h. After 24 h, the culture produced detectable foci of infection. Infection with *Salmonella typhimurium* led to the presence of bacteria in the organoid lumen. Authors suggest using bovine organoids in further epizootic/epidemic research of other pathogens.

Powell et al. also isolated sheep ileum-derived organoids in 2017 [56]. The cultures were established from four donors and successfully achieved 66 passages (239 days of growth). LGR5 RNA expression was higher in the organoids than in the control tissue, and low vimentin expression (mesenchymal marker) suggested that mesenchymal cells are not necessary for the expansion of the organoids. This observation is aligned with the situation in other veterinary species.

Abomasal and ileal organoids were derived by Smith et al. in 2021 as a model to study host-pathogen interactions [60]. The organoids were derived from 7–8-month-old Texel cross male lambs using protocols for intestinal organoid culture in farm animals described by Hamilton et al., 2018 [120] and Beaumont et al. in 2021 [102]. The authors investigated the invasive capabilities of a whole parasite (*Teladorsagia circumcincta*) in organoid culture. Twenty-four hours after parasite introduction, the parasite burrowed into the Matrigel and inside the organoids. Motile worms were present in the culture 14 days after their introduction to the culture. The authors experimented with polarity reversal and successfully created apical-out abomasal and ileal organoids by incubating the organoids in 5 mM EDTA for 1 h and removing the Matrigel, an approach described by Co et al. in 2019 [121]. The authors concluded that parasitic pathogens could invade organoids and that organoids might be used for host-pathogen interaction modeling. Further investigation here is warranted.

The first equine ileal organoids were described in 2017 [56] and were derived from three horses with a 100% success rate using 50% L-WRN-conditioned media. The highest passage number for these cultures was 46, representing 168 days in culture. The authors used Miyoshi and Stappenbeck’s previously described murine isolation protocol, published in 2013 [118]. Low vimentin expression in the organoids compared to the tissue suggested that the organoids could grow without the addition of mesenchymal cells.

The isolation of small intestinal rabbit organoids (duodenum, ileum, colon) from three adult European rabbits (*Oryctolagus cuniculus*) was reported by Kardia et al. in 2021 [61]. While the media supported the growth of all three intestinal parts harvested from a laboratory rabbit, only duodenal organoids could be derived from wild-caught rabbits using L-WRN-conditioned media. The authors used a modified version of the murine intestinal organoid protocol published by Miyoshi and Stappenbeck in 2013 [118]. The most efficient growth was observed with an L-WRN-conditioned medium supplemented with ROCKi and a TGF-β inhibitor. The authors achieved the growth of the duodenal enteroids in a monolayer system in a traditional culture plate and a Nunc chamber slide system. A lower concentration of L-WRN-conditioned media with an addition of ROCKi and DAPT (Notch pathway inhibitor) and removal of TGF-β inhibitor was used to promote organoid differentiation. A functional test was performed at the culture consisting of inoculation of Rabbit calicivirus Australia-1 to the organoid culture with a negative result. The authors proposed that their culture could have lacked sensitive cell types, or co-factors might have been required to produce an infection.

The first feline ileal organoids were developed by Powell et al. in 2017 from nine donors [56]. The success rate of organoid growth was less than 50%, and the highest passage number achieved was 18 (67 days in culture). Feline ileal organoids seemed to cease their expansion around passage 10 and arrest their growth around passages 13 to 18. The culture before passage 10 included mesenchymal cell types that disappeared in further passages and might be important for organoid expansion and growth. This problematic culture condition led the authors to perform a separate experiment to identify the proper composition of the organoid media for feline organoids. The tested components included A83-01 (ALK5 inhibitor), SB202190 (p38 MAP kinase inhibitor), human FGF-4, 2, and 10, nicotinamide, human IGF-1, prostaglandin E2, mouse Wnt-2b, and human Gremlin. None of the mentioned components saved feline organoids. Authors suggested using feline organoids to study host–pathogen interactions, for example, the interaction with *Toxoplasma gondii*. Vimentin is expressed in levels above that of the control tissue in feline organoids, which suggests the presence of mesenchymal cells in organoid cultures, further expanding the hypothesis that feline ileal organoids need an additional type of cell to expand and grow. Protocol adjustments are warranted for further experiments.

Tekes et al. in 2020 investigated the best practices for the long-term culture of feline ileal and colonic organoids to investigate feline coronaviruses, including feline enteric coronavirus (FECV) and deadly feline infectious peritonitis virus (FIPV) [62]. The established organoid culture was characterized by qPCR, showing the presence of all intestinal types. Notably, colonic tissue did not express a marker for Paneth cells (Lyz). The optimal passaging method for feline intestinal culture was investigated using three methods (trypsinization, mechanical disruption, and gentle cell dissociation reagent). Mechanical disruption seems to be the ideal method for bringing the highest yield of organoids with the highest organoid diameter.

Another experiment was performed to compare the growth of the organoids in traditional human and mouse media. In both the ileum and colon, the cell lines grew more successfully in human media. The cell cultures from both organs could not support the growth of FIPV, and ileal organoids could not support FECV growth. Feline colonic organoids can support FECV infection and, therefore, might be used for further investigation of feline coronaviral infections.

### 4.3. Liver Organoids

The significance of organoids in disease modeling is predicated on the fact that biopsies from individual patients can be used to construct disease-specific organoids. In addition, a putative disease-causing mutation can be repaired, allowing for cause-and-effect research. The unnatural culture conditions with excess growth factor(s), oxygen, and other nutrients, as well as the lack of interaction with other cell types (stem cell niche), require caution when comparing in vitro and in vivo findings.

The first healthy canine hepatic-derived organoids were reported by Nantasanti et al. in 2015 [66] and were used as a model for gene therapy in Wilson’s disease. While this disease does not occur in dogs, its pathophysiology can be mimicked in COMMD1-deficient dogs, causing toxic copper accumulation in the liver. The scientists note that the hepatic organoids mainly expressed stem or progenitor cell characteristics, needing better hepatocyte differentiation methods. The work describes the formation of canine (liver) organoids and presents proof that the genetic abnormality causing COMMD1 deficiency and subsequent hepatic copper buildup was effectively repaired by lentiviral transduction of the complete coding sequence of the COMMD1-gene. These gene-corrected autologous hepatic stem cells were subsequently transplanted into recipient COMMD1-deficient dogs via the portal vein and the animals survived for at least 2 years; however, following engraftment in the liver, hardly any proliferation in these cells was observed, and there was no functional recovery with respect to copper accumulation and biliary copper excretion [122]. Organoids generated from healthy livers and livers from dogs with congenital portosystemic shunts (CPSS) were used to collect lipids if grown in the presence of excess free fatty acids in the medium and mirrored the steatosis frequently reported in CPSS livers [123]. As another example of disease modeling, canine liver organoids from healthy and COMMD1-deficient dogs and repaired COMMD1-deficient organoids were found to exhibit decreased FXR transcriptional activity [124], which is consistent with data from copper-laden human livers.

Further research on canine hepatic organoids was described in 2022 [30], with the standardization of canine hepatic and intestinal cultures. Adult-derived canine organoid technology lacks standardized protocols for handling procedures due to its relative novelty; therefore, this is a barrier for intra-laboratory and inter-laboratory research comparisons. Canine organoid protocols were based on the work of Saxena et al., 2015 [112], studying rotaviral infection in human intestinal organoids and a follow-up protocol by Zou et al., published in 2019 [125], exploring human intestinal organoids for rotaviral and noroviral infections.

The author’s group described three individual protocols encompassing organoid (1) culture establishment, (2) growth, and (3) organoid harvest. (1) Culture establishment is described in the Canine Organoid Isolation Protocol, ranging from obtaining the sample to embedding the tissue suspension. The (2) Canine Organoid Maintenance protocol describes the general upkeep of the organoid culture and procedures ensuring good growth of the culture. These procedures include organoid cleaning (removal of necrotic or apoptotic cells) and organoid passaging (dissolving organoids into smaller cellular clusters to expand the established culture). Lastly, Gabriel et al. introduced the (3) Canine Organoid Harvesting and Biobanking Protocol describing how to prepare the organoid culture for downstream applications, including fixing and paraffin-embedding, liquid nitrogen archival and revival of the organoids, and the preservation of RNA from the organoid samples.

Feline hepatic organoids were developed and characterized by Kruitwagen et al. in 2017 as a model to study hepatic steatosis [67]. The source of five feline hepatic biopsies included fresh, frozen, and fine-needle aspirates. The culture achieved a passage number of 25. The organoid culture had stem cell characteristics (gene expression of LGR5, PROM1, and BMI1). Genes of cholangiocyte/progenitor cells (KRT7, KRT19, and HNF1β) and early hepatocyte markers (HNF4α and TBX3) were also expressed in the culture, whereas mature hepatocyte-specific markers (PROX1, PC, HMGCL, TTR, FAH, and CYP3A12) were not expressed. Organoids were cultured in differentiation media (nicotinamide, R-spondin 1, forskolin, and Y27632 withdrawal; BMP7, DAPT, dexamethasone addition) to increase the number of hepatocyte-like cells. The cells that underwent differentiation accumulated glycogen (positive PAS staining), had increased aspartate aminotransferase (AST) levels, and produced albumin in the media.

The gene expression of mature hepatocyte markers was significantly increased compared to that in the expansion media control, while stem cell marker (LGR5) expression dramatically decreased. The proliferation of the culture abruptly stopped after the introduction of the differentiation media. Free fatty acids (oleate and palmitate) were supplemented to the culture media to mimic conditions of hepatic steatosis, and the organoids were found to accumulate more lipid droplets than that in the human organoid control.

The interference of small molecules with feline hepatic organoids was tested to validate the model. Etomoxir disrupted fatty acid metabolism and increased free fatty acid accumulation. Conversely, L-carnitine attenuated free fatty acid accumulation, as previously reported, and ameliorated free fatty acid oxidation in cats with lipidosis. The authors concluded that feline-derived organoids could be a valuable model for hepatic steatosis research due to felines’ predisposition for lipid accumulation.

Haaker et al. continued the study of feline hepatic organoid applications in 2020, using the technology to identify potential novel drugs for feline hepatic lipidosis treatment [126]. Two drug candidates were identified based on a free fatty acid accumulation experiment similar to that by Kruitwagen et al., 2017. The drugs decreased triacylglycerol accumulation in differentiated (hepatocyte-like) and undifferentiated (cholangiocyte) hepatic organoids. Therefore, the authors argue that differentiation may not be an important first step, and the screening of undifferentiated organoids might also be useful.

### 4.4. Renal Organoids

The Madine-Darby canine kidney (MDCK) cell line is arguably the most commonly utilized canine cell line. In contrast to this well-known cell line, which is essential for research on cellular polarization, there is only one publication on canine kidney organoids [45]. The authors focus on the role of organoids in the biomedical research of end-stage renal disease and establishing proper culture conditions (by the inclusion of N-acetylcysteine, ascorbic acid 2-phosphate, nicotinamide, and FBS). Furthermore, Chen et al. compared their culture with the MDCK 2D traditional culture established in 1958 [127]. Once cultured in Matrigel, multipotent canine kidney cells with mesenchymal stem cell (MSC)-like characteristics (adipogenic, chondrogenic, and osteoblastic differentiation) developed tubule-like structures. Although CD24 and CD133 expression is consistent with previously defined renal ASCs, the MSC-like phenotype is more indicative of an MSC than an adult kidney stem cell. Renal ASCs grown in Wnt-stimulating media display CD24 and CD133 but have no adipogenic, chondrogenic, or osteoblastic differentiation capacity [45].

### 4.5. Pancreatic Organoids

Investigation into pancreatic organoids can lead to discoveries in the physiology and pathophysiology of the organ. An interesting study was published by Liu et al. in 2021 [46] investigating the influence of copper supplementation on the ovine pancreas. Copper supplementation was crucial for organoid growth; a similar finding was previously described in another in vivo experiment, where copper deficiency led to ataxia in newborns and unspecific symptoms in adults [128]. The authors isolated pancreatic ducts from a fetal sheep and processed the samples based on a rodent primary pancreatic cell isolation protocol by Agbunag et al. published in 2006 [129].

The pancreatic organoids showed the characteristics of pancreatic epithelial progenitors with additional stem cell markers (LGR5 and EPCAM). No expression of endocrine cell markers was revealed (insulin, glucagon, and amylase); therefore, the organoids were dubbed “sheep pancreatic duct organoids” due to the high expression of cytokeratin 19 (a ductal marker). The addition of RSPO1 and EGF to organoid media was essential, as more cyst-like organoids were observed in supplemented media than in the control. This finding was confirmed with the EdU proliferation assay and CCK8 viability assay. The authors also successfully used a forskolin-induced swelling assay to prove CFTR activity and an assay to measure chloride movement to confirm the functional properties of the organoids.

An experiment with the addition of the copper chelator tetrathiomolybdate to the medium was performed, and the inhibition of organoid formation and growth was observed and confirmed with a CCK8 assay. The proper dose of copper supplementation also enhanced the growth of the organoids (10 µM). However, supplementation with excessive amounts of copper (100 µM) significantly inhibited the growth of organoids. Both findings were confirmed using EdU and CCK8 assays. The authors also identified a pathway responsible for the effects of copper on organoids (MEK-ERK pathway) and further investigated the copper interaction using a knockdown organoid model.

### 4.6. SARS-COVID-2

The SARS-COVID-2 pandemic prompted investigation into coronaviral infections, and two model organoid lines were developed to facilitate investigation into the disease. Bat enteroids were derived by Zhou et al. in 2020 [64] using a protocol for deriving human intestinal organoids published by Sato et al. in 2011 [32]. The author’s main area of interest was a study of the infection caused by SARS-COVID-2 in an intestinal bat model. The interest was based on the first research at that time proposing a connection of the pandemic to a horseshoe bat (*Rhinolophus sinicus*) host [130]. Zhou et al. showed that bat enteroids are susceptible to COVID-19 infection and that the virus can replicate in organoids. This discovery provided evidence of SARS-CoV-2′s ability to infect bat intestinal cells. The authors acknowledge that they did not achieve long-term organoid expansion as described in Sato’s protocol, and that further protocol optimization will be needed. Finally, Zhou et al. noted that numerous viruses with zoonotic potential are present in the bat intestine, and bat organoids can be a useful tool for studying them.

In 2021, Elbadawy et al., described the establishment of intestine-derived organoids from Leschenault’s rousette fruit bat (*Rousettus leschenaultia*) [65]. The authors identified proper culture conditions for a long-term expansion of the culture. An experiment was performed showing that Wnt, R-Spondin, Noggin (collectively called WRN), transforming growth factor alpha (TGF-α), and epidermal growth factor (EGF) were the most important media factors to sustain organoid growth and proliferation. The intestinal organoids were not susceptible to SARS-CoV-2 but were sensitive to Pteropine orthoreovirus (PRV) inoculation. Elbadawy et al. further stated that bat intestinal organoids could be used to investigate other rousette bat-associated viruses, such as Ebola or Marburg virus.

### 4.7. Precision/Personalized Medicine

Elbadawy et al., further investigated utilization of the organoid technology for precision medicine purposes in patients with bladder cancer [42]. The authors’ data suggest that their research could be used as a potential canine bladder cancer treatment. More information on the use of personalized and precision medicine in the field of veterinary medicine was already described in a publication by Mealey et al. from 2019 [131]. 

### 4.8. Other Applications

#### 4.8.1. Reproductive Physiology and Pathology

Animal-derived organoids were also developed to investigate reproductive organ physiology and disease pathophysiology. The lines encompass organoids derived from the endometrium and oviducts. Equine uterine organoids (endometrium) were derived in 2020 by Thompson et al. [63] from biopsies of 11 adult domestic mares (*Equus ferus caballus*) and 3 Przewalski’s mares (*Equus ferus przewalskii*). Equine endometrial organoids produced mucin, which was detected using (PAS) staining. Organoids were then tested for an exogenous hormonal response.

The organoids from a Przewalski horse were the first organoids from an endangered species. This significant observation demonstrates that it is possible (and thus novel) to study biological processes in endangered species without sacrificing the animals. Importantly, prolactin promoted the synthesis of beta-casein in equine mammary organoids [43], which was described functionally. Although horses are not commonly employed for milk production, this large animal model may permit cross-species comparisons to better comprehend the lactation mechanism.

Estradiol-17β reduced ESR1 gene expression in frozen and fresh tissue-derived organoids alike. PGR gene expression was increased in fresh tissue-derived organoids. While the organoids reacted to hormonal changes, it was in unexpected ways, without resemblance to the in vivo reaction. The authors argue that part of the feedback mechanism for gene regulation might be missing, or the hormonal diffusion through the extracellular matrix might be decreased. Furthermore, Thompson et al. published research on cryopreservation discussing the organoid growth rate, cellular viability, and other properties of equine endometrium-derived organoids [132].

Thompson et al. continued their research on equine organoids by adding an equine oviductal organoid protocol [133]. The culture was established for 42 days and included three passages. Further investigation into this model is warranted. The authors propose using organoid technology to advance artificial reproductive technique research.

Feline endometrial organoids exhibited polarization and laminin expression in the basement membrane [134]. Since this organoid system was created to assess the toxicity of plastics or as a possible source of novel antibiotics, no information on cellular differentiation was supplied.

#### 4.8.2. Skin Diseases

The production of scar tissue in the skin as a result of damage has social consequences and impairs quality of life, and this fact has prompted research into skin organoids and the development of canine skin organoids [68]. Samples were harvested from twelve 1- to 2-year-old donors euthanized for other reasons. The authors used microdissection methods previously described elsewhere [135] to isolate hair follicles and the interfollicular epidermis. The two samples were then treated separately for all dogs. The follicles and epidermis were digested with collagenase 1A and II and hyaluronidase and subsequently trypsinized overnight to obtain a single-cell keratinocyte suspension. The sample was strained, and cells were seeded in Matrigel.

Immunohistochemistry and gene expression profiling confirmed that stem cell-based organoids were proliferative in the expansion medium (Wnt stimulation and BMP inhibition) and differentiation medium (i.e., neither R-spondin nor Noggin), with a shift in keratin family member expression indicating the expression of typical layered skin cell types. While the expansion media included R-spondin-1 and Noggin, surprisingly, Wnt3a, FGF1, and FGF18 media supplementation did not improve organoid growth.

Most organoid cultures derived from follicles or the epidermis had a finite time of expansion ranging from six to fifteen passages. However, the authors also described one organoid line that survived for 50 passages (independent of the presence of A-83-01—a TGF-β inhibitor). Both types of organoids had a similar morphology, as observed by microscopy (H&E) of paraffin-embedded samples. The authors describe the presence of the analogs of individual squamous epithelial layers, including stratum basale, spinosum, granulosum, and even stratum corneum. Some of the organoids encompassed keratinized lumen or keratohyalin-like granules.

Immunohistochemistry revealed that KRT5 (basal layer marker) was expressed in the outer layer of the organoids, KRT6 (early differentiation marker) was highly expressed in the center of the organoids, and Ki67 (proliferation marker) was highly expressed in the outer layer. The expression levels of these markers were confirmed by qPCR and RNAseq.

Intriguingly, lgr5 mRNA was found only in hair follicle (HF) tissue and not in organoids, whereas lgr6 mRNA was expressed in both HF and intrafollicular epidermis (IFE) organoids in early passages. This indicates distinct stem cell pools in the mouse HF isthmus, IFE, and sebaceous glands [68]. To further improve canine skin organoids which can represent the various epidermal cell layers, the culture medium was modified [136]. The most unexpected discovery was that IFE-derived organoids grown in expansion medium (EM) mirrored the layered skin architecture. 

#### 4.8.3. Ophthalmology

Recent progress using stem cell technology was also noted in the field of veterinary ophthalmology. Bedos et al. developed for the first time corneal organoids from canine and feline tissue in 2022 [69]. The key corneal epithelial and stromal cell markers were detected in the organoids using RNA in situ hybridization method. To the author’s knowledge, this was the first time corneal organoids were isolated from adult stem cells in any species. While the experiment was successful in creating organoid structures, further investigation is warranted to improve growth conditions of the culture.

#### 4.8.4. Antivenom Production

Antivenom production depends on access to snake venom, harvested via “venom milking”. The process requires an experienced herpetologist to force a venomous snake to bite down on a plastic membrane covering a collection jar to stimulate venom ejection from fangs and subsequent harvesting of the product. Venom extraction and snake housing methods have slowly improved over the past century, and Grego et al. offer an informative review on the topic mapping the history of the Brazilian Butantan Institute’s venomous snake housing since 1908 [137]. This harvested venom is used to immunize experimental animals (mostly horses); blood or plasma is subsequently collected, and the active immunoglobulin substances are purified [138] (Figure 4). The antivenom production process is labor and cost intensive with various yields and, therefore, low antivenom supply security [139]. The mortality level associated with snake bites has been estimated to be approximately 125,000 deaths per year, and the high demand for antivenom treatment warrants novel solutions [140].

In 2020, Post et al. described organoids derived from the venom glands of nine snake species [41]. These organoids produced venom peptides reflecting the composition of crude venom and its biological activity. Snake-derived organoids could be used to harvest venom, a necessary component in antivenom manufacturing. If expanded to a sufficiently large scale, this technology could replace venom collection via venom milking, a dangerous and technically difficult process.

This research is also unique in establishing the first organoids from the class Reptilia. Post et al. confirmed that Wnt signaling bears the same importance in reptile organoid growth as it does for organoids derived from mammalian species. This observation was also supported by the need to add the Wnt pathway agonist R-spondin to the media to establish organoid lines successfully. The authors suggest that generally established organoid growth conditions might apply to all vertebrate species. Puschhof et al., also from the Hans Clevers group, published a detailed protocol for isolating snake venom gland organoids for the in vitro production of snake venom in 2021 [38].

## 5. Summary

Access to new and robust in vitro biomedical models is warranted to promote medical research. Significant progress has been made in the field of organoid research. Rodent organoids can be used to investigate the genetic origins of disease, but in many areas, the translatability of the model is low. Harnessing the unique adaptations seen across the animal kingdom benefits biomedical advances and translational research. While many of these animal-derived organoids can be used in unique biomedical applications, the canine model can be specifically utilized in a reverse translational manner to promote knowledge in both human and veterinary medicine.

## Figures and Tables

**Figure 1 ijms-25-00701-f001:**
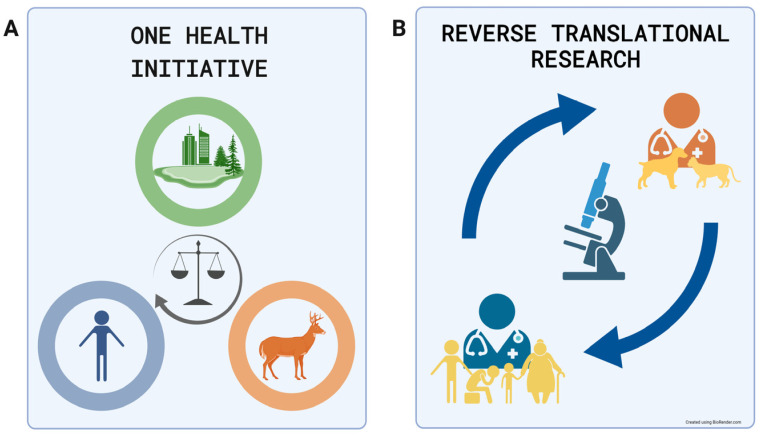
One Health Initiative and Reverse Translational Research. (**A**) The One Health Initiative is a novel approach that seeks unified solutions for human, animal, and environmental health issues and improves communication between biomedical scientists. The approach aims to improve biomedical sciences and form novel research strategies. (**B**) Reverse translational research, as a part of the One Health Initiative, promotes cooperation and knowledge sharing between human and animal clinicians and clinical/paraclinical researchers. The concept is based on sharing clinically relevant patient data wherever possible for translational purposes.

**Figure 2 ijms-25-00701-f002:**
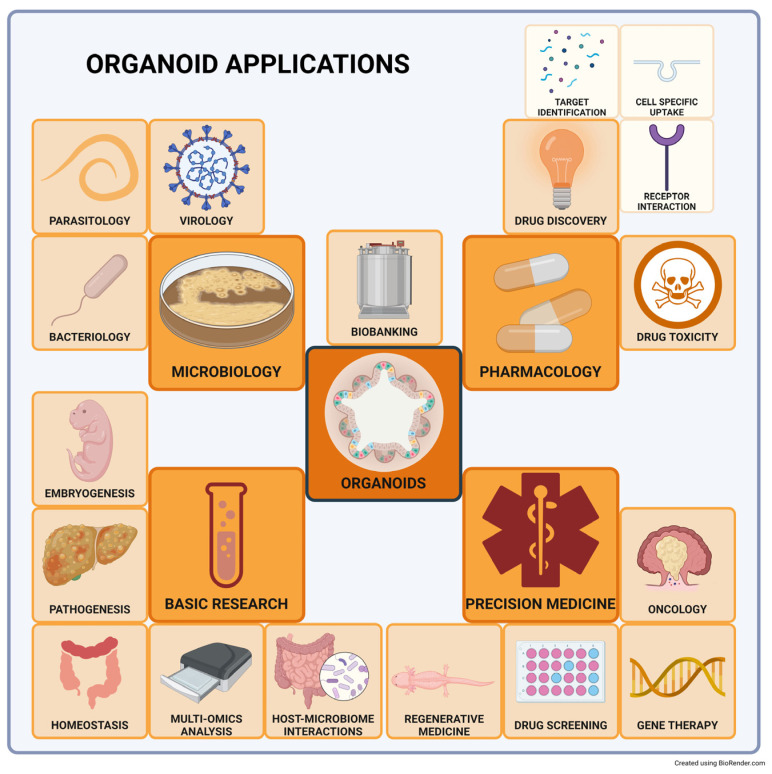
Organoid applications. Organoids can be used in numerous fields within biomedical research. Dark orange frames represent the main topics, light orange frames represent subfields, and white frames are the detailed applications.

**Figure 3 ijms-25-00701-f003:**
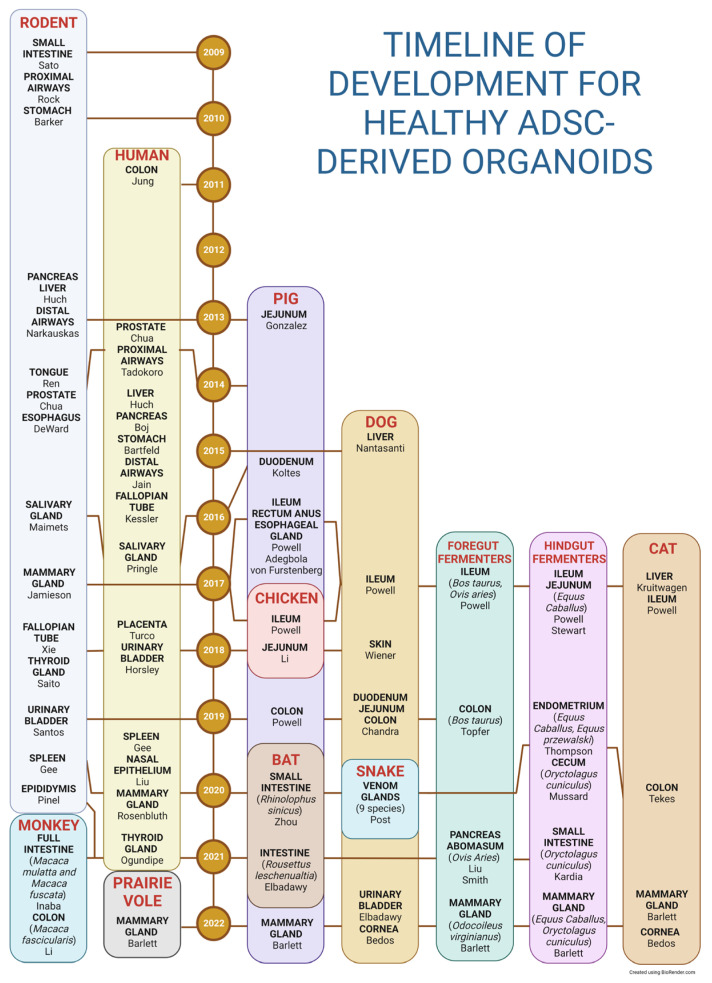
Timeline of Development for AdSC-derived Organoids. The timeline represents the first description of healthy organoid development in humans, rodents, and numerous other species, including traditional veterinary species and exotic animals.

**Figure 4 ijms-25-00701-f004:**
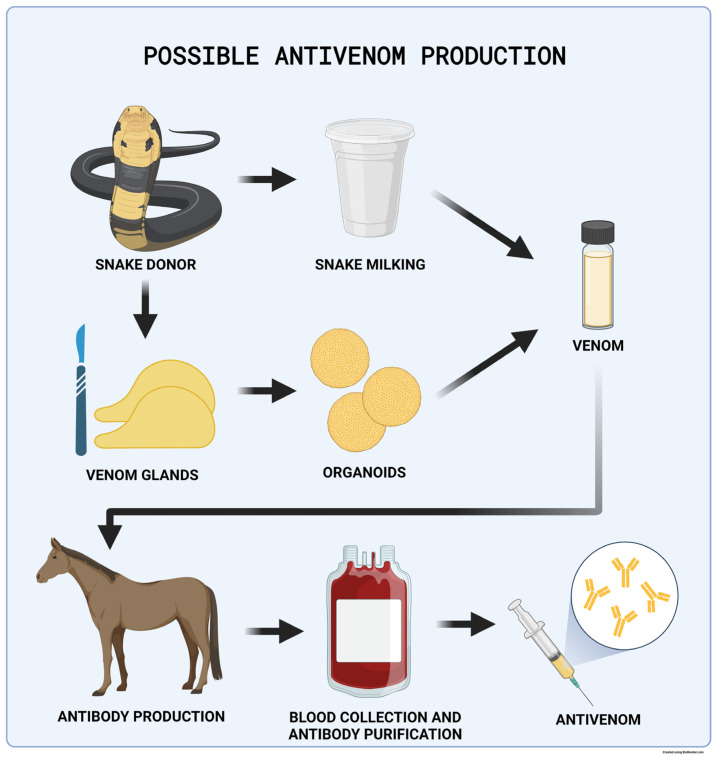
Possible antivenom production process. Snake venom gland organoids could be used as an alternative to laborious snake milking. The venom could be introduced into an animal, and the immunized animal could produce antibodies. The animal’s blood would be collected, and antibodies could be purified for the final product—antivenom.

**Table 1 ijms-25-00701-t001:** Overview of the first isolations of nonrodent and nonhuman animal-derived organoids as biomedical models. A list of healthy organoids including their species, the article’s author, the organ of origin, and their proposed utility in biomedical research.

Biomedical Model For	Species	Taxonomical Name	Author	Organ/Intestinal Segment
Antivenom Production	Red spitting cobra	*Naja pallida*	Post et al. 2020 [41]	Venom Gland
	Snouted cobra	*Naja annulifera*		
	Cape cobra	*Naja nivea*		
	Chinese cobra	*Naja atra*		
	Cape coral snake	*Aspidelaps lubricus cowlesi*		
	West African carpet viper	*Echis ocellatus*		
	Chinese moccasin	*Deinagkistrodon acutus*		
	Western diamondback rattlesnake	*Crotalus atrox*		
	Puff adder	*Bitis arietans*		
Urinary Bladder Cancer Model	Dog	*Canis lupus familiaris*	Elbadawy et al. 2022 [42]	Urinary Bladder
Mammary Gland Cancer	Prarie Vole	*Microtus ochrogaster*	Barlett et al. 2022 [43]	Mammary Gland
	Pig	*Sus scrofa domesticus*		
	White-tailed deer	*Odocoileus virginianus*		
	Horse	*Equus ferus caballus*		
	European rabbit	*Oryctolagus cunniculus*		
	Cat	*Felis catus domestica*		
Drug Permeability and Toxicity Testing	Crab-eating Macaque	*Macaca fascicularis*	Li et al.2021 [44]	Colon
	Dog	*Canis lupus familiaris*	Chen et al. 2019 [45]	Kidney
Inflammatory Bowel Disease			Chandra et al. 2019 [23]	Intestine
Influence of Copper on Pancreas	Sheep	*Ovis Aries*	Liu et al. 2021 [46]	Pancreas
Intestinal Physiology Investigation	Rhesus Macaque	*Macaca mulatta*	Inaba et al. 2021 [47]	Full Intestine
	Japanese Macaque	*Macaca fuscata*		Full Intestine
	Chicken	*Gallus gallus domesticus*	Li et al. 2018 [48]	Jejunum
	Cattle	*Bos taurus*	Topfer et al. 2019 [49]	Colon
	Horse	*Equus ferus caballus*	Stewart et al. 2017 [50]	Jejunum
	European rabbit	*Oryctolagus cunniculus*	Mussard et al. 2020 [51]	Cecum
	Pig	*Sus scrofa domesticus*	Gonzalez et al. 2013 [52]	Jejunum
			Adegbola et al. 2017 [53]	Anus, Rectum
			von Furstenberg et al. 2017 [54]	Esophageal Glands
Inflammation Signaling and Edotoxine Tolerance	Pig	*Sus scrofa domesticus*	Koltes et al. 2016 [55]	Duodenum
Pathogen-Host Interaction	Pig	*Sus scrofa domesticus*	Powell et al. 2017 [56]	Ileum
	Chicken	*Gallus gallus domesticus*		
	Pig	*Sus scrofa domesticus*	Li et al. 2019 [57]	Colon
	Dog	*Canis lupus familiaris*	Powell et al. 2017 [56]	Ileum
			Acharya et al. 2020 [58]	Small Intestine
	Cattle	*Bos taurus*	Derricott et al. 2019 [59]	Jejunum
			Powell et al. 2017 [56]	Ileum
	Sheep	*Ovis Aries*		
			Smith et al. 2021 [60]	Abomasum
	Horse	*Equus ferus caballus*	Powell et al. 2017 [56]	Ileum
	European rabbit	*Oryctolagus cunniculus*	Kardia et al. 2021 [61]	Duodenum
				Ileum
				Colon
	Cat	*Felis catus domestica*	Powell et al. 2017 [56]	Ileum
			Tekes et al. 2020 [62]	Colon
Reproductive Mechanisms	Horse	*Equus ferus caballus*	Thompson et al. 2020 [63]	Endometrium
	Przewalski’s horse	*Equus ferus prezewalski*		
	Horse	*Equus ferus caballus*		Oviduct
SARS-COVID-2	Chinese rufous horseshoe bat	*Rhinolophus sinicus*	Zhou et al. 2020 [64]	Small Intestine
	Leschenault’s rousette	*Rousettus leschenualtia*	Elbadawy et al. 2021 [65]	Intestine
Wilson’s Disease	Dog	*Canis lupus familiaris*	Nantasanti et al. 2015 [66]	Liver
Hepatic Steatosis	Cat	*Felis catus domestica*	Kruitwagen et al. 2017 [67]	
Epidermal Function and Cutaneus Disorders	Dog	*Canis lupus familiaris*	Wiener et al. 2018 [68]	Skin
Ophthalmology	Dog	*Canis lupus familiaris*	Bedos et al. 2022 [69]	Cornea
	Cat	*Felis catus domestica*	Bedos et al. 2022 [69]	Cornea

## Data Availability

Not applicable.

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
