# Peer review of "Adult Animal Stem Cell-Derived Organoids in Biomedical Research and the One Health Paradigm"

_ijms, 2024, doi:10.3390/ijms25020701_

Round 1

Reviewer 1 Report

Comments and Suggestions for Authors

This is a comprehensive review were systematically the various veterinary relevant organoids are being described. The addition of the venom glands isa novelty, to my knowledge, not previously addressed in review papers. The extended introduction on why organoids is placed in one health paradigm which is a powerful approach.

The manuscript would be further strengthened if at place where the authors, usually at the end of a paragraph, refrase the conclusion of the paper discussed. It is not alway clear how the conclusion by the referred authors is substantiated by their data or is a generalisation or even more concerning a hopeful believe. As reader of a review this more critical assessment of referred papers would be appreciated. On the other hand the extended reference list provides the readers sufficient links to make up their own mind on those papers.

some minor points:

line 319: CHIT is indeed a GSK-3 inhibitor, but in the context of Wnt signaling its consequential Wnt stimulatory role seems more relevant for the line of this manuscript

line 359 most likely in vivo the polyamines come form the intestinal bacteria

line 515 presence esp for efflux punps is not equal to activity (which is easy measurable) 

line 566 why a kidney cell line for intestinal research?

Author Response

Dear Reviewer 1, thank you very much for your comments. We appreciate you finding time in your schedule to review this manuscript, this has been one of the fastest reviews we have experienced with our other manuscripts so far and a very positive experience.

When preparing the manuscript, the point from the second paragraph has been on our mind several times and I completely agree with your conclusions. I went through the manuscript and added a few sentences to some generalized statements from the original literature because I also believe some of them are too broad. We tried to solve this situation by paraphrasing the conclusions of the authors. The situation here is very unique due to the novelty of organoid/stem cell technology, especially in veterinary medicine. It is very hard to distinguish between what authors are sure about based on their internal experimentation and what is an overarching conclusion supported by human/rodent research. We know that the stem cell research field will grow exponentially in the upcoming years and most of the organoid research will have to be tested, expanded on, and confirmed by other research groups. Either way, this should be clearly pointed out by our side, and we improved on that in the manuscript.

Some minor points:

line 319: CHIT is indeed a GSK-3 inhibitor, but in the context of Wnt signaling its consequential Wnt stimulatory role seems more relevant for the line of this manuscript

I changed the sentence in the manuscript to connect Wnt and GSk-3 inhibition. Thank you very much for the comment

line 359 most likely in vivo the polyamines come form the intestinal bacteria

Thank you for the comment, I added the information in the manuscript.

line 515 presence esp for efflux punps is not equal to activity (which is easy measurable) 

Thank you and I added a sentence regarding possible additional measurements of the efflux pump activity in the manuscript.

line 566 why a kidney cell line for intestinal research?

This is a very good question; I added an explanation of why the cell line was used for this purpose to the manuscript. It always surprises me what kind of cell lines were used before the establishment of organoid technology. Thankfully, organoids can provide more precise physiological and pathophysiological models.

Dear Reviewer 1, thank you again for all your comments, we very much appreciate you work on our manuscript. Happy Holidays to your family!

Reviewer 2 Report

Comments and Suggestions for Authors

Thank you for this very informative and interesting paper. The review entitled “Animal Adult Stem Cell-Derived Organoids in Biomedical Research and the One Health Paradigm”. The aim of this review was to demonstrate the benefits of using animal adult stem cell-derived organoids for biomedical research. Despite the many reviews on this topic, the authors were able to find and reflect their point of view on this problem. This is an interesting review, and the authors have collected a sufficient dataset using the related methodology. The review is well written, structured, and illustrated. In my opinion, it is necessary to add a section to the section, Reproducibility and data deposition, in order to reflect the problems that exist today and ways to overcome them.

Author Response

Dear Reviewer 2, first, let me thank you for your speedy review of our manuscript. Your work is greatly appreciated. Secondly, thank you for the kind words, and I am very glad this is your opinion about our manuscript. Adding a Reproducibility and data deposition section would be a great idea. In a time when veterinary organoid research looks much more like “Wild West,” we should focus on standardizing our cultures to ensure reproducibility and improve the means/culture of public data deposition to support our conclusions. I added the Reproducibility and data deposition section to the manuscript. Thank you again for your help, and I wish you Happy Holidays!

Reviewer 3 Report

Comments and Suggestions for Authors

The manuscript entitled “Animal Adult Stem Cell-Derived Organoids in Biomedical Research and the One Health Paradigm” is a review article that is overall, informative and well-written. I have only a few comments regarding some topics that could be expanded upon a bit.

Page 6, Line 190 – This section explains the isolation of Lgr5+ cells as an intestinal multipotent stem cell marker but there is very little description of the isolation of the adult stem cells from other tissues. Although the actual organoids are the focus of this review, it would be helpful to briefly touch on how the adult stem cells used to make these are isolated from key tissues. Are similar cell surface markers used for isolations across these tissues? Different markers? Generally, the same across species?

Page 7 – Table 1 resolution is low and blurry on the copy I am reviewing.

Page 7, line 253-258 – This particular paragraph has a writing style that is less clear than the rest of the manuscript – “plentiful groups of organoids” do you mean many? Also “Mouse-derived organoids usually represent the first translational model of choice for producing human organoids” not clear what this means.

General – Some limitations of using adult stem cell derived organoids were not discussed:

1)      Based upon this review – it seems these are only feasible when studying certain tissues – not neurological or striated muscle (skeletal or cardiac muscle), others.

2)      What are the limitations regarding ability to maturate these different organoids? For any of the applications reviewed in the manuscript - what developmental stage is really being modeled? Do people know how mature their organoids are, how significant is it depending upon the application?

Author Response

Dear Reviewer 3,

Thank you very much for the fast review of our manuscript. We acknowledge the time you spent on our manuscript, and we are grateful for your help. Also, thank you for the kind words, and we appreciate your comments regarding our manuscript!

Page 6, Line 190 – This section explains the isolation of Lgr5+ cells as an intestinal multipotent stem cell marker but there is very little description of the isolation of the adult stem cells from other tissues. Although the actual organoids are the focus of this review, it would be helpful to briefly touch on how the adult stem cells used to make these are isolated from key tissues. Are similar cell surface markers used for isolations across these tissues? Different markers? Generally, the same across species?

Thank you, that is an interesting question, and I appreciate it very much. I added a shorter paragraph with an explanation and linked standardized protocols describing the isolation of various canine tissue types.

Page 7 – Table 1 resolution is low and blurry on the copy I am reviewing.

Thank you. We are adding a table with higher resolution; we will also make high-resolution figures available to the journal.

Page 7, line 253-258 – This particular paragraph has a writing style that is less clear than the rest of the manuscript – “plentiful groups of organoids” do you mean many? Also “Mouse-derived organoids usually represent the first translational model of choice for producing human organoids” not clear what this means.

I changed the wording in the paragraph to make more sense from a reader’s perspective; thank you for the catch!

General – Some limitations of using adult stem cell derived organoids were not discussed:

1)      Based upon this review – it seems these are only feasible when studying certain tissues – not neurological or striated muscle (skeletal or cardiac muscle), others.

Thank you for this comment, we added a few sentences to explain the difference between AdSCs and iPSCs and the possibilities for aforementioned disease/physiology investigations.

2)      What are the limitations regarding ability to maturate these different organoids? For any of the applications reviewed in the manuscript - what developmental stage is really being modeled? Do people know how mature their organoids are, how significant is it depending upon the application?

Thank you for this comment. This is a good question, and Reviewer 2 asked us to add the Reproducibility and Data Deposition section to our manuscript. The developmental stage is another variability I am adding to this section.

Thank you again for improving our manuscript, and I wish you a Happy Holidays!
